# A Low-Power Analog Processor-in-Memory-Based Convolutional Neural Network for Biosensor Applications

**DOI:** 10.3390/s22124555

**Published:** 2022-06-16

**Authors:** Sung-June Byun, Dong-Gyun Kim, Kyung-Do Park, Yeun-Jin Choi, Pervesh Kumar, Imran Ali, Dong-Gyu Kim, June-Mo Yoo, Hyung-Ki Huh, Yeon-Jae Jung, Seok-Kee Kim, Young-Gun Pu, Kang-Yoon Lee

**Affiliations:** 1Department of Electrical and Computer Engineering, Sungkyunkwan University, Suwon 16419, Korea; steven7264@skku.edu (S.-J.B.); exeric@skku.edu (Y.-J.C.); itspervesh@skku.edu (P.K.); imran.ali@skku.edu (I.A.); rlarlarbrb@skku.edu (D.-G.K.); fiance2@g.skku.edu (J.-M.Y.); hara1015@skku.edu (Y.-G.P.); 2SKAIChips, Suwon 16419, Korea; horsnal@skku.edu (D.-G.K.); gray@skaichips.co.kr (H.-K.H.); yjjung@skaichips.co.kr (Y.-J.J.); skkim@skaichips.co.kr (S.-K.K.); 3Department of Artificial Intelligence, Sungkyunkwan University, Suwon 16419, Korea; pkd0213@skku.edu

**Keywords:** convolutional neural network, processor-in-memory, AI controller, smart sensing controller, on-chip implementation on a biosensor

## Abstract

This paper presents an on-chip implementation of an analog processor-in-memory (PIM)-based convolutional neural network (CNN) in a biosensor. The operator was designed with low power to implement CNN as an on-chip device on the biosensor, which consists of plates of 32 × 32 material. In this paper, 10T SRAM-based analog PIM, which performs multiple and average (MAV) operations with multiplication and accumulation (MAC), is used as a filter to implement CNN at low power. PIM proceeds with MAV operations, with feature extraction as a filter, using an analog method. To prepare the input feature, an input matrix is formed by scanning a 32 × 32 biosensor based on a digital controller operating at 32 MHz frequency. Memory reuse techniques were applied to the analog SRAM filter, which is the core of low power implementation, and in order to accurately grasp the MAC operational efficiency and classification, we modeled and trained numerous input features based on biosignal data, confirming the classification. When the learned weight data was input, 19 mW of power was consumed during analog-based MAC operation. The implementation showed an energy efficiency of 5.38 TOPS/W and was differentiated through the implementation of 8 bits of high resolution in the 180 nm CMOS process.

## 1. Introduction

As an artificial intelligence model, CNN is considered to be suitable for classification [1,2]. It is common to classify the images that are input through photos or image sensors after dataset-based learning [3,4]. Since each neuron makes classifications based on the features of input data, identifying the features between data is the key to CNN operation. For example, in a model for image recognition, each neuron is divided into neurons based on red, green, and blue for input data. Neurons with high values for pixels with red, neurons with high values for pixels with green, and neurons with high values for pixels with blue have high values. In the case of biosensors, when a disease is included, neurons related to the disease have high values. In recognition of this CNN model’s excellent classification ability, it is also applied to the field of disease classification through the use of a biosensor [5,6]. Prior to the application of the CNN model, scientists relied strongly on the data accumulated from the experience of researchers and analysis of detailed differences between the data. However, CNN-based biosensor systems can obtain the desired data from unlabeled data through learning. The CNN approach is being used to conduct excellent research, not only in image classification, but also in biosensor applications [7,8].

However, there are many restrictions associated with using CNN in biosensor applications: (1) memory for computation, (2) energy efficiency, (3) bottleneck phenomenon, and (4) timing analysis [9,10]. Due to the above limitations, it is not easy to apply a typical CNN structure to biosensor applications and implement it as an SoC. To overcome the above limitations, research is being conducted in two main directions to integrate the current CNN algorithm into an on-chip system. These two types of research include an ASIC-based accelerator and an accelerator using a high-performance FPGA. First, ASIC-based algorithmic research proposes a structure that satisfies energy efficiency, throughput, etc. [11]. Second, algorithm research based on FPGA is actively being conducted, including work on parallel operation and desired interface configuration [12].

A neuromorphic chip refers to a chip made using technology that mimics the synapse structure of a person’s brain [13,14]. Unlike the CPU method, which is a traditional limited parallel or sequential processing, in this method, neurons that imitate the human brain are configured in parallel, and they have excellent scalability and excellent learning ability in a small area. Aside from the amount of data, the computational performance is maintained [15]. Existing CPU-based operations constantly generate power consumption while processing data between CPU and memory. However, in the case of a neuromorphic chip, it plays a decisive role in minimizing energy consumption, as it is frequently disconnected when working or not working, similar to a brain synapse. An artificial intelligence-based SoC receives data from the sensor and creates input data for CNN operation through a sensing and processing block that extracts only the desired information from the sensor data. To provide the desired output data to the user through the layer in which many neurons exchange data and memory, the designers have the duty to implement it with ultra-low power.

The motivation of this work is to design a CNN algorithm with ultra-low power to apply to biosensor applications. This paper describes the on-chip CNN design method, which is distinguished by obtaining biosignal information from the biosensor, processing the signal, and classifying the desired information using a typical AI algorithm using a computer [16], Memory allocation is important when using artificial intelligence algorithms in a computer, but on-chip, how much and how efficiently all the memory required for calculations is used is the key. Our design applied analog-based PIM to operate MAC filtering in RTL-based CNN architecture and CNN operations [17], PIM stores weight, advances MAC operations in parallel, and enables fast calculation. Further, the analog-based filter can be reused for each layer of operation and used for the overall operation in one analog filter structure, without any additional memory allocation for the filter [18], Sharing and using filters, regardless of the layer, leads to good performance, not only in terms of power consumption, but also in terms of area. The proposed analog-digital hybrid CNN has been applied to classify biosensor data through parallel operations. The proposed technology enables immediate utilization in pandemic situations or new disease outbreaks. There is no need to spend much time developing the kit, and new diseases can be detected with only detection information about the disease.

The rest of this paper proceeds as follows: Section 2 introduces a low-power CNN top architecture that is applicable to biosensor applications and FSM-based controllers that are implemented digitally. It also introduces the detailed computational behavior and detailed architecture of an analog filter based on analog 10T SRAM, which is used as an important filter for computation. Section 3 describes the performance of the architecture’s software and hardware and introduces the advantages of the proposed structure; measurement results and a comparison with other work are also included. Finally, the conclusion of the paper is summarized in Section 4.

## 2. Architecture

### 2.1. Top Architecutre

Figure 1 is a CNN top architecture implemented on-chip in the biosensor. First, the biosensor consists of a 32 × 32 based material plate consisting of four layers. When the sample is dropped on the biosensor, the amount of current changes. The amount of change in current depends on the type of disease contained in the sample. To diagnose a disease by detecting the amount of current change in the 32 × 32-based biosensor, the current change is detected in the analog front-end, and the current information is transferred to the digital block through the ADC. To diagnose a specific disease using the data of the biosensor as an input feature, CNN was applied on-chip with a biosensor. ASIC’s CNN algorithm trains the amount of change in current according to the disease and identifies the disease when the sample is dropped on the biosensor. For diagnosis, learning is required based on information about the disease, and the learning is applied to ASIC by extracting weight information from the CNN model through TensorFlow. On-chip implementation, including a GUI interface, is used to check disease outcome. Low-power operations were implemented through analog PIM to apply CNN on-chip to the biosensor. Analog PIM basically stores the filter’s value on the CNN and performs MAC operations using feature extraction. Because of this process, there was no need for additional memory allocation for MAC operation, and the operation function was not required, so it could be implemented with low power.

### 2.2. Detailed Structure

#### 2.2.1. Input Feature

Figure 2 shows the Input Feature Map input after detecting the information from the biosensor with the analog front-end. The data is converted to digital code through ADC from a biosensor application and then input to a CNN block. This is a model that classifies 10 classes, from disease A to disease J, and if a disease is applicable, then only a specific part of the input feature has a high code, whereas the other part has a relatively low code. The reason why the area is divided according to the disease is that antibodies against different diseases are accumulated in the biosensor. When the virus is dropped, a change occurs in response to the antibody in the corresponding part. For example, as shown in Figure 3, if disease A is included, the digital code in a specific part of the biosensor retrieves 700 to 800 codes based on the 16 bit ADC. Conversely, relatively few codes, e.g., 0–200, are retrieved in areas that do not contain diseases. With these features, an input feature map is created that uses information sensed by the biosensor as CNN input to classify the disease. Through the modeling input feature map, the CNN architecture is first configured as a code level, and learning is conducted using the final actual data.

#### 2.2.2. The Proposed CNN Model Architecture

Figure 4 shows the proposed CNN model. The proposed model consists of a total of six layers to be implemented on-chip and applied to the biosensor. It consists of two convolution layers, two max pooling layers, and fully connected and softmax output layers. The first layer performs a convolution operation with kernel size 4 × 4 as the input feature and uses the ReLU function as the activation function. Then, in the pooling layer, the size of 29 × 29 is halved to 14 × 14, with a 2 × 2 size stride 2 to optimize the operation. The third and second convolution operations perform convolution operations on 14 × 14 layers with a kernel size of 4 × 4, which is the same as in the first layer, and they use the ReLU function as the activation function. Then, they reduce the 11 × 11 input feature map to a size of 5 × 5 with stride 2, with a 2 × 2 size as the fourth layer. After that, all matrices are sorted into vectors to form a fully connected layer connecting all neurons. The softmax layer, which classifies the class with the highest probability as the last output layer, was composed of size 10. Analog 10T SRAM-based PIM was used as a filter, MAC operations were performed through a capacitor inside the PIM structure when calculating with feature extraction, and low-power operations were implemented to enable on-chip application by reducing data movement between unnecessary operators and memory.

#### 2.2.3. The Proposed Analog CNN Filter

Figure 5 shows the structure of the analog processor-in-memory filter in the CNN architecture: It consists of an SRAM of 16 × 4, 16 DACs, and 4 ADCs. A controller is included to optimize the timing of SRAM, DAC, and ADC operations and enable and disable operations to reduce current consumption; and there is also a main controller that manages the timing and operation of all controllers. The input for computation is from the AI Controller to the DAC Controller, since DAC uses a charge sharing structure. The low power implementation was optimized by restricting all blocks to operate only when needed.

Figure 6 shows the detailed structure of SRAM and ADC/DAC. A 4 × 4 filter, consisting of an SRAM memory cell and an MAV operator, forms four channels in total. ADC and DAC have a charge sharing structure to minimize current consumption. Weight is updated to the SRAM cell of the analog PIM core through the AI controller. Upon completion of the weight update, the input feature and the weight proceed with the MAV operation. The MAC operation is the same as shown in Equation (1). k refers to a column, $Y_{out}$ is the result of the operation through the final ADC, and $X_{in}$ is the input feature. The MAV operation is based on the operation of dividing the weight and input data by N after multiplication.
(1)$$Y_{out,k}=\frac{1}{N}\underset{i}{\sum }W_{k,i}\times X_{IN,i}$$

Both the multiplication and the average of the weight and input feature are composed of analog operations. A capacitor was used to perform analog operations, and the basic synthesis equation in the serial connection of the capacitor for understanding MAC operations through the capacitor is expressed in Equation (2).
(2)$$\frac{1}{C}=\frac{1}{C_{1}}+\frac{1}{C_{2}}+\frac{1}{C_{3}}=\frac{C_{1}\times C_{2}\times C_{3}}{C_{2}C_{3}+C_{3}C_{1}+C_{1}C_{2}}$$

If $C_{1},\;C_{2},\;\mathrm{and}\;C_{3}$ are all the same capacitance, then Equation (2) may be expressed as Equation (3).
(3)$$C=\frac{C_{a}{}^{3}}{3C_{a}{}^{2}}=\frac{C_{a}}{3}$$

Therefore, an analog-based average calculator was implemented by applying a method of obtaining synthetic capacitance by connecting capacitors having the same capacitance in series. The equation of the MAV operation is as follows: BL refers to the bitline in SRAM, and the Nth bitline and weight calculate each other. After that, the final convolution result is derived by dividing it by N.
(4)$$V_{conv}=\frac{\left(V_{BL1}\times W_{1}\right)+\left(V_{BL2}\times W_{2}\right)+\cdots +\left(V_{BLN}\times W_{N}\right)}{N}$$

When the input feature is input to the DAC controller, the DAC controller generates a clock pulse and outputs a DAC voltage that increases linearly by the number of clock pulses. ADC also makes the output voltage of the analog PIM core a clock pulse and compares it with the reference voltage that increases linearly with its number, which causes the number of clock pulses to become ADC output data.

Figure 7 shows that the output data after MAV operation using one row at a time PIM and the output of the voltage charger—which is linearly charged over time—are input to the ADC comparator. Counting the number of clock pulses until the moment the voltage of the voltage charger exceeds the output voltage of the PIM core, it is then exported to the ADC output. The higher the resolution, the larger the size of the capacitor in the charge sharing structure; in this process, the offset occurs due to the mismatch between the SRAM cell and capacitor size. Calibration logic was implemented inside the ADC to compensate for the offset, and the offset was calibrated to derive an accurate value.

## 3. Experimental Results

### 3.1. Implementation

Figure 8 shows the CNN controller’s layout. The AI smart sensing controller (AISSC) and the AI neuromorphic controller (AINC) are composed of synthetic RTL digital code, and the synthesis and place and route (PnR) processes are performed to configure analog block and merge. The AISSC layout size is 1350 × 1400 ${\mathrm{um}}^{2}$, and the AINC is implemented as 1720 × 1650 ${\mathrm{um}}^{2}$, which is a slightly larger layout.

Figure 9 shows the layout of the analog processor-in-memory, which includes DAC, SRAM, ADC, and the controller. This was implemented using the 180 nm CMOS process and the performance was optimized with the most symmetric deployment. Figure 10 shows a die photograph of the SoC. The current consumption values of ADC, DAC, and SRAM are 113.2 uA, 284.2 uA, and 50 nA, and the current consumption values of the SRAM controller, ADC controller, DAC controller, and the main controller are 16 uA, 423 uA, 225.1 uA, and 279.4 uA, respectively.

### 3.2. Classification

Figure 11 shows the RTL simulation results for CNN full path behavior. The final output class was confirmed through the operation of convolution layers and max pooling. The output class was 10, and the input data was modeled and simulated. The operation time in the process of receiving and calculating input data is 15 ms. Updated weight data was extracted as a file through CNN simulation modeled with TensorFlow, then used for simulation. It was confirmed that the output result was suitable for the class.

### 3.3. CNN

#### 3.3.1. CNN Simulation

Figure 12 and Figure 13 show the simulation results for the CNN full path. Figure 12 shows the simulation results of the AINC, which reads data from SRAM and performs the convolution layer and max pooling layer operations. As each layering operation proceeds, the feature map for each layer is read/write, and an FSM-based convolution operation is implemented. The final output class was 10, and it was confirmed through simulation that the output class was successfully determined according to the range of input data. Figure 13 shows the simulation results of the enabled feature map and the write/read data, which performs and stores convolution operations from an input feature map in an AI neuromorphic controller. Since the operation is performed in parallel with four filters, the feature mapping from IFM to FM3 is performed simultaneously. CNN parallel operations implemented on-chip have the advantage of being able to derive fast response speeds with the same power consumption; the key is to write during other layer operations. Accordingly, the operation speed may be improved and stabilized through management of the timing issue and parallel processing.

#### 3.3.2. AI Main Controller (AIMC)

Figure 14 shows the simulation result for the AI main controller inside the AISSC block. Basically, AISSC is the top module of all controls, such as scanning the biosensor, communicating with other controllers, and communicating with the GUI. Among these, AIMC consists of an algorithm that receives data from the sensor and delivers it to the logic, which creates input data. Instead of always receiving data through the controller, the operation is started by receiving data when an operation is required.

#### 3.3.3. Communication and Control Interface

Figure 15 shows the slave controller simulation results of the internal module of the AISSC for the communication and control interface. This is a controller for receiving control signals, input data, or weight data from the GUI to verify behavior and check output. It has the major function of delivering data corresponding to frames. The FDATA value is transferred in series and controlled so that the value can be checked in the GUI.

### 3.4. Processor-in-Memory

#### 3.4.1. PIM Controllers

Figure 16 shows a PIM controller that is in charge of controls such as timing and calibration for the implementation of low power analog PIM. To operate the PIM in the AI controller, the main controller must first be given an enable signal. When the main controller receives an enable signal, it first charges the DAC through the DAC controller, then reports that it has been charged through the EOC signal after operation, and finally enables the SRAM to operate. After SRAM operation, the EOC signal is also transmitted to the main controller and an enable signal is transmitted to the ADC controller so that the ADC can operate. Each operation of DAC, SRAM, and ADC is only turned on when an operation is needed, and they are not in an always on state. In Figure 16b, one can see the overall signal flow in the timing diagram. It is proposed for a low-power SoC implementation method by controlling the detail.

#### 3.4.2. PIM Core

Figure 17 shows the result of simulating the sequential operation from the analog processor in memory to the DAC-SRAM-ADC. The above graph shows the DAC operation, and the lower graph shows the operation of ADC. It was confirmed that charging was performed up to 600 mV at 2.34 mV per step, with an 8-bit resolution. After delivering data through ADC, refresh is performed for the next operation; this represents a 1 cycle operation.

### 3.5. Measurement Result

#### 3.5.1. Measurement Environment

Figure 18 shows the measurement environment set up to measure CNNs based on the analog processor-in-memory. Based on the computer GUI, we measured the performance of the model by entering the input data and the weight on the AI board through FPGA. FPGA delivers data from the GUI to the biosensor to control the SoC, implementing a PIM-based CNN Model.

#### 3.5.2. CNN Classification

Figure 20 shows the results of verifying the CNN full path after checking for the filter result of the analog PIM. The weight of the learned model should be extracted, prepared as a text file, and input into the GUI for testing. Figure 19 shows the data required as the input feature. There is an antibody in the biosensor, which generates a change in current when the antibody reacts with a disease, and by detecting this change, it is necessary to determine which part of the 32 × 32 material substrate has the highest value. The CNN model classifies classes by learning where and by how much the value has increased. When B disease is dropped on the substrate, it leads to a high current change in the position, as shown in Figure 19. When the test is conducted with the B disease input data, it can be seen that the class is classified accurately, as shown in Figure 20. Input data and weight were entered in the form of a txt file, and the operation was confirmed by applying a 1.8 V voltage.

#### 3.5.3. Processor-in-Memory

This is the result of the proposed analog PIM-based hybrid CNN model SoC measurement, as opposed to a typical CNN structure. After implementing SoC, a board was produced and measured, and Figure 21 shows the measurement result that verifies the analog PIM behavior through the GUI. The PIM operation is performed to verify whether the MAV operation is accurate when any input data and weight are entered. The input data and weight of filters 1 to 4 obtained the same results as the measurement results from AINC_CONV 0 to 3, with the ideal value obtained by MAV operation, according to Equation (4). However, although the offset was reduced through calibration logic, the offset of about 0 to 2 could be confirmed, and when the offset was confirmed for various cases, as shown in Figure 22, it was confirmed that there was an offset of about 0 to 2. However, the offset of 0 to 2 is an offset allowance that can be sufficiently removed through CNN operations. The analog PIM-based hybrid CNN model was applied to the biosensor and implemented for the purpose of solving memory allocation for the AI model’s common problem, operation, and improving energy efficiency by performing operations without additional functions. As a result of the measurement, the energy consumption during MAC operation is 19 mW, which shows improved energy consumption compared to the general structure. It was implemented with low power and showed peak energy efficiency of 5.38 TOPS/W at 32 MHz. It was also possible to improve the bottleneck phenomenon between the operator and the memory by performing the operation in memory.

Table 1 is a table comparing the performance of the proposed model with that of other works by applying artificial intelligence technology for diagnosing diseases to the on-chip method. A smaller area can be realized while the microprocessor is used, and the issues that occur when a higher CLK is used are predictable. In the case of CLK, since it has a large effect on the CNN operation speed, it is advantageous for a model that requires more computations, as it is faster. Moreover, the lower the voltage, the higher the efficiency in terms of energy consumption. In the case of resolution, if high resolution is used, energy consumption or operation speed may decrease, but high accuracy can be derived. The present work was conducted using a relatively larger process (180 nm) than that used in other works, and a 1.8 V voltage domain was used, but it showed excellent performance for MAC energy consumption. It was designed to use 32 MHz by adjusting the clock of PIM for low power implementation, and the internal ADC was designed to use 8 bits, and it was optimized.

## 4. Conclusions

This paper presents a low-power analog processor-in-memory-based CNN algorithm for biosensor applications. In artificial intelligence computation using FPGA or a computer using GPU and CPU, there are many limitations in interworking with the biosensor and constructing it as a kit. Unlike artificial intelligence models, implemented based on FPGAs and computers, when implementing on-chip, it is necessary to design an operator that considers memory and power consumption, and both operation resolution and interfaces for learning and checking results must be considered. In this paper, all things considered, on-chip processing was implemented as a low-power hardware SoC through MAV (multiple and average) operation using an analog PIM. Further, a system was implemented to classify the data received from the biosensor by applying the light CNN model.

The proposed analog processor-in-memory-based CNN structure was implemented using a 180 nm CMOS process, and the size of the processor-in-memory was 2000 × 600 ${\mathrm{um}}^{2}$. The MAC energy consumption showed a performance of 19 mW at 1.8 V supply. Further, the energy efficiency was 5.38 TOPS/W, thus showing excellence in the analog-digital hybrid CNN structure. The ADC resolution was also implemented at high resolution by showing 8 bits of performance in a relatively large process of 180 nm. Altogether, we proposed an algorithm that can detect diseases by accurately classifying the data received from a biosensor. Based on this study, it has been proven that diagnosis is possible by utilizing artificial intelligence implemented using a biosensor and on-chip processing, and it is expected that rapid diagnosis will be possible for new and more complex diseases.

## Figures and Tables

**Figure 1 sensors-22-04555-f001:**
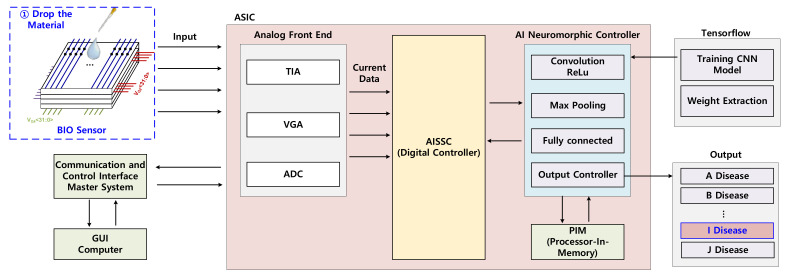
Top Architecture of Analog-Digital Hybrid Convolutional Neural Network in a Biosensor.

**Figure 2 sensors-22-04555-f002:**
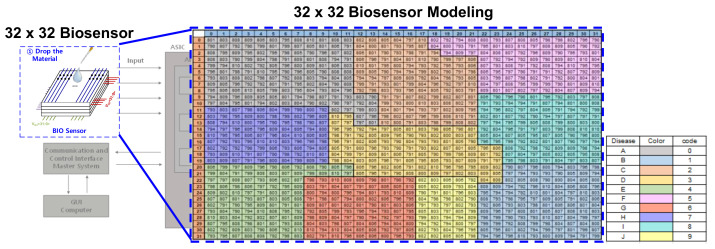
Input Feature Map of CNN. ADC output data extracted through the front-end block by receiving input from the biosensor.

**Figure 3 sensors-22-04555-f003:**
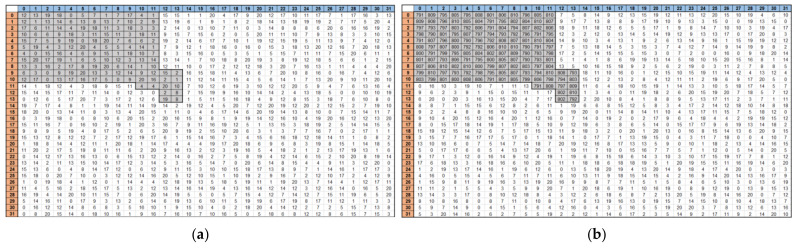
Comparison of biosensor modeling (**a**) without and (**b**) with diseases.

**Figure 4 sensors-22-04555-f004:**
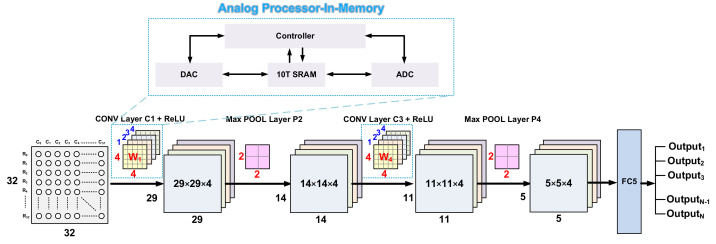
The Proposed CNN Model Architecture.

**Figure 5 sensors-22-04555-f005:**
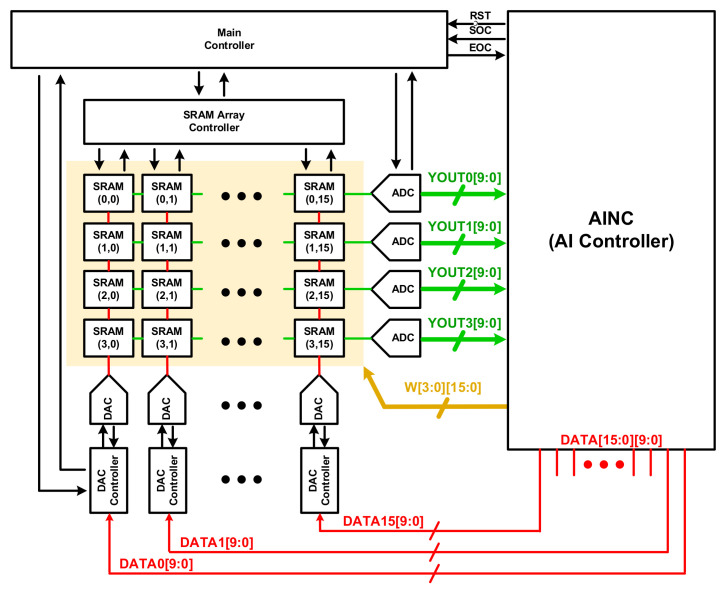
The Proposed Analog Processor-In-Memory of CNN.

**Figure 6 sensors-22-04555-f006:**
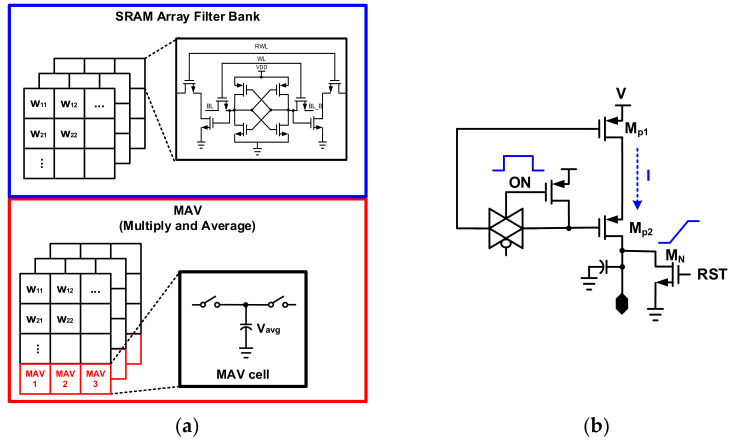
A detail structure of the analog processor-in-memory: (**a**) 10T based 4 × 4 size SRAM cell and MAV operation; (**b**)voltage charger with charge sharing structure used in DAC/ADC.

**Figure 7 sensors-22-04555-f007:**
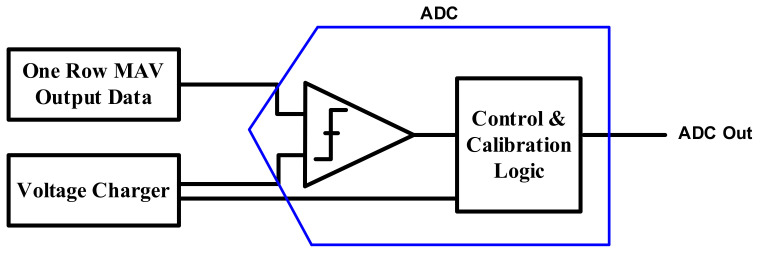
The proposed ADC architecture of analog PIM with calibration.

**Figure 8 sensors-22-04555-f008:**
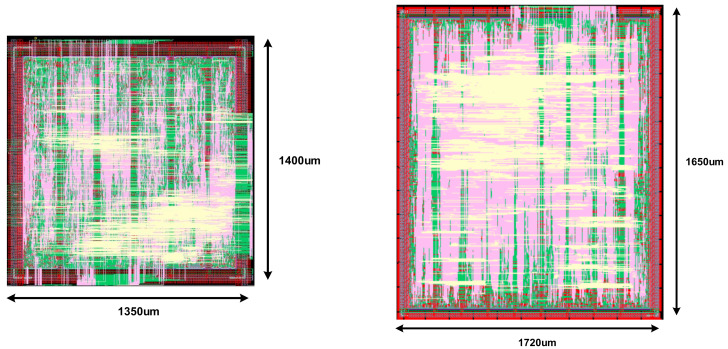
The layout of the controller of the CNN: (**a**) AISSC layout; (**b**) AINC layout.

**Figure 9 sensors-22-04555-f009:**
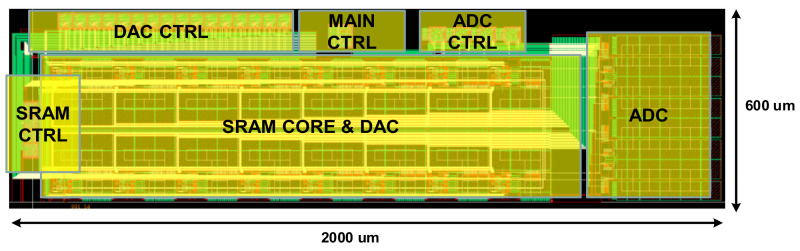
Processor-In-Memory Layout.

**Figure 10 sensors-22-04555-f010:**
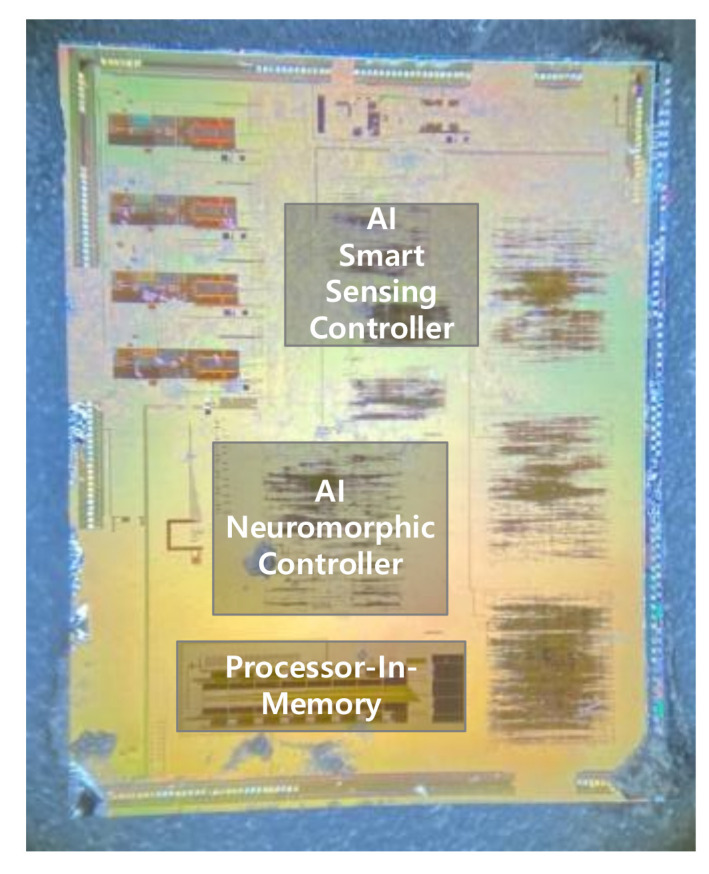
Die Photograph in a 180 nm CMOS Process.

**Figure 11 sensors-22-04555-f011:**
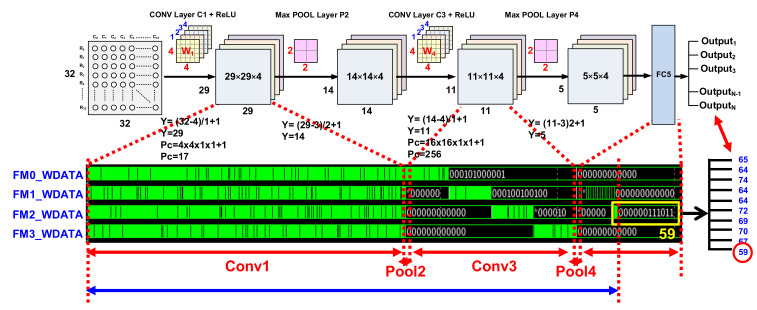
The simulation result of CNN architecture.

**Figure 12 sensors-22-04555-f012:**
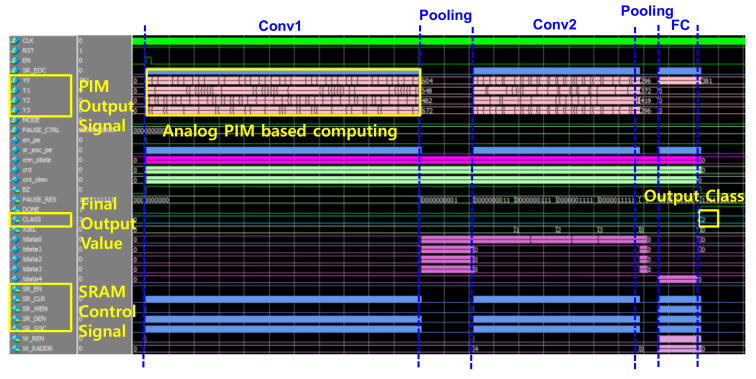
The simulation result of top CNN includes PIM to analyze classification.

**Figure 13 sensors-22-04555-f013:**
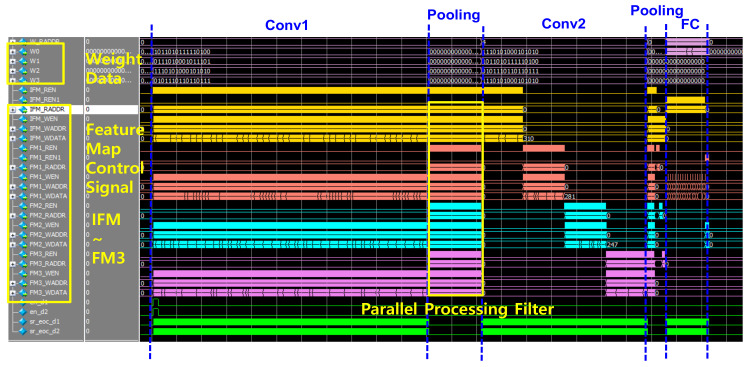
The simulation result of top CNN to analysis feature map.

**Figure 14 sensors-22-04555-f014:**
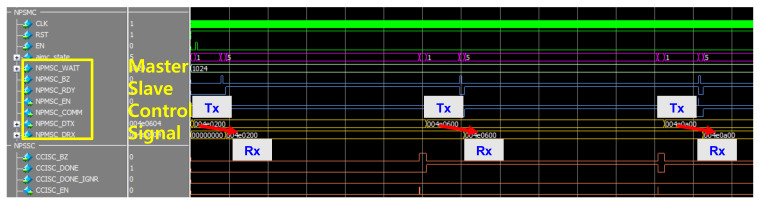
The simulation result of the AIMC control for the biosensor.

**Figure 15 sensors-22-04555-f015:**
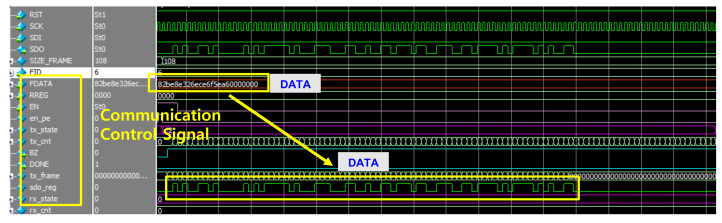
The simulation results of communication and control interface.

**Figure 16 sensors-22-04555-f016:**
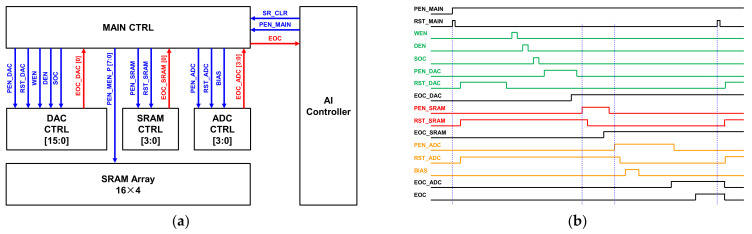
Detailed results of the analog PIM controller: (**a**) block diagram between controllers; (**b**) timing diagram.

**Figure 17 sensors-22-04555-f017:**
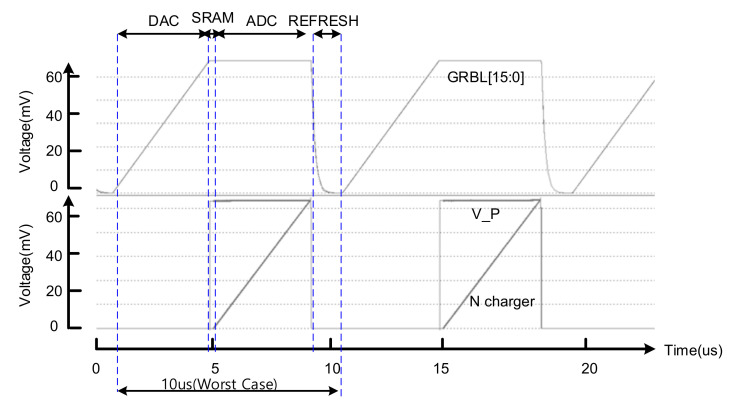
The simulation results of the analog processor-in-memory core.

**Figure 18 sensors-22-04555-f018:**
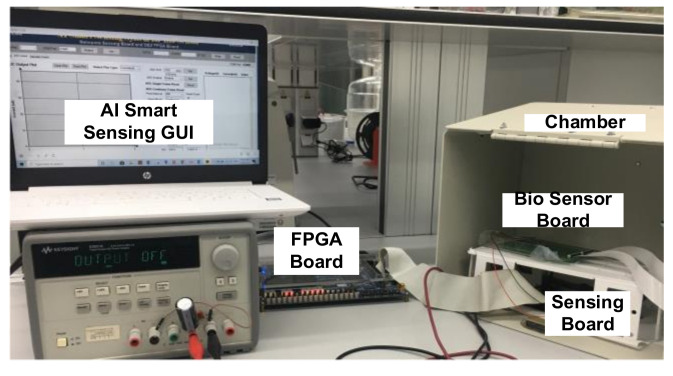
Measurement setup on Board with FPGA.

**Figure 19 sensors-22-04555-f019:**
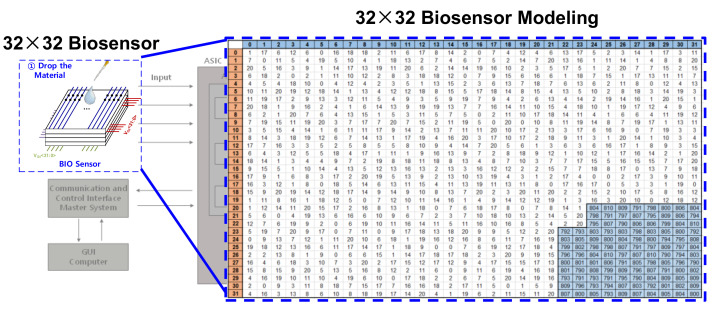
Input feature of B disease.

**Figure 20 sensors-22-04555-f020:**
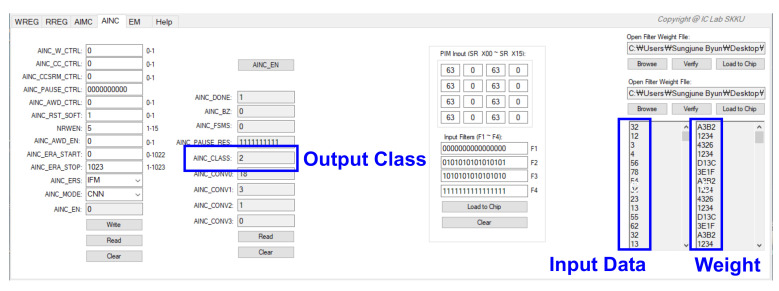
Measurement result of CNN Classification.

**Figure 21 sensors-22-04555-f021:**
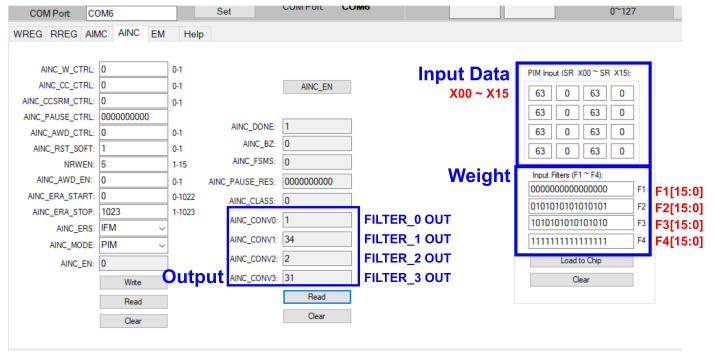
Measurement results of filter output.

**Figure 22 sensors-22-04555-f022:**
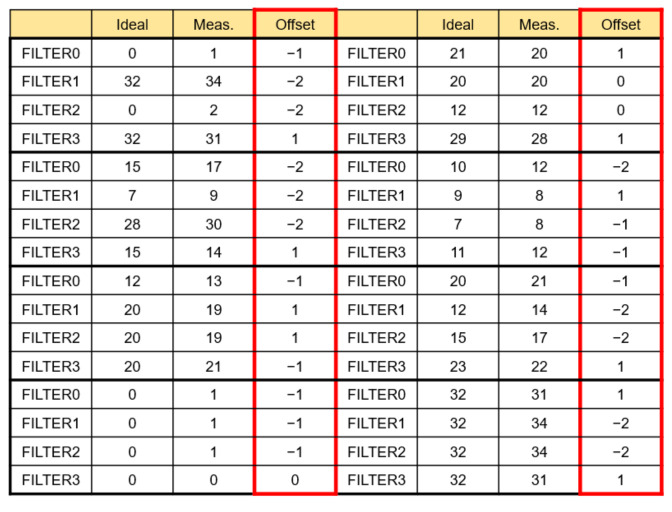
Measurement results of filters output offset.

**Table 1 sensors-22-04555-t001:** Performance Comparison.

Parameter	This Work	[19]	[20]	[21]	[22]
Process (nm)	180	130	130	130	65
Architecture	Analog/Digital Mixed Processor-In-Memory	Analog/Digital Mixed	Analog/Digital Mixed	Digital	Digital
Clock (MHz)	32	200	200	200	110~400
		10			
Voltage (V)	1.8	1.0/1.2	1.2	0.65/1.2	0.55~1

Resolution (bits)	8	4	8	-	-
MAC Energy Consumption (mW)	19	75	496	260	50–600
Peak Energy Efficiency (TOPS/W)	5.38	0.65	0.29	0.646	2.3

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
