# Peer review of "A Low-Power Analog Processor-in-Memory-Based Convolutional Neural Network for Biosensor Applications"

_sensors, 2022, doi:10.3390/s22124555_

Round 1
Reviewer 1 Report
In this work is constructed and implemented in a Bio Sensor an analog Processor-In-Memorybased Convolutional Neural Network .
In the introduction the authors need to highlight more the performance advantages of their research compared to other works in the literature where CNN in biosensor application are implemented.
A more in-depth discussion of the architectural model in Fig. 1 is needed.
The performance comparison results in Tab. 1 must be fully discussed. The values of all the parameters seem very distant from those measured using the approaches in [18,19,20]. Authors must justify these findings.
A discussion of the limitations, advantages and future prospects should be added in the conclusions.
Author Response
First of all, thank you for your review.
The introduction you mentioned emphasized the performance benefits of the study more compared to other references.
I added discusion for Fig.1
I added specific description for table 1.
I added advantages and future prospects to the conclusion.
Reviewer 2 Report
This paper proposes an on-chip implementation of an analog Processor-In-Memory (PIM)- based Convolutional Neural Network (CNN) in a Bio Sensor. It was implemented as low power hardware SoC through multiple and average (MAV) operations using analog PIM. Further, a system was implemented to classify the data received from the biosensor by applying the light CNN Model. This network has certain superiority and effectiveness. This is an interesting research paper. There are some suggestions for revision.
1) The motivation is not clear. Please specify the importance of the proposed solution.
2) Please highlight the innovations of the proposed solution.
3) At the end of the abstract, it is more intuitive and convincing to illustrate the qualitative results of a large number of experiments for verifying the superiority and effectiveness.
4) In the first paragraph of Section 1, the paper mentions that “Since each neuron makes classifications based on the features of input data, identifying the features between data is the key to CNN operation”. Please specify the data characteristics.
5) The explanation of equations 1-4 is not enough, What the symbol on the right side of the equation represents puzzles the readers.
6) In the second paragraph of Section 3.5, Figure 19 which is needed as the input feature ought to be clearer in order to display the content more intuitively.
7) The technical depth of the proposed solution is weak. More technical details should be given.
8) The experimental results are not convincing. Please compare the proposed solution with more recently published solutions.
9) The conclusion does not further discuss the advantages and weaknesses of the design, and the significance of the research and the prospects for the future are not sufficiently proposed.
10) Most of references are a little bit out of date. Please discuss more recently published solutions.
11) English grammar errors and inaccurate descriptions cause confusion and make interpretation difficult to understand. The authors are advised to correct the linguistic errors in the edited text.
Author Response
First of all, thank you for your review.
I emphasized the importance of the paper in the introduction part.
In addition, I wrote specifically about the proposed solution in the conclusion.
I modified it to make it more clear about abstract.
I added the details of what you said in the introduction.
We explained the formula in detail and modified it to make it easier for the reader to understand.
I added more details to make it a little clearer about Figure 19.
I added it as a whole because it seemed unclear what should be considered in the typical CNN and On-Chip implementations (low power, operation memory allocation, and understanding of the hardware perspective).
I added a recent reference to the table and configured it for comparison with a recent reference.
I added advantages and future prospects to the conclusion.
I also added a reference to the recent data to compare with the recent paper.
I checked the English grammar.
Round 2
Reviewer 1 Report
Authors took into account all my suggestions. I consider this paper publishable in the present form.
Reviewer 2 Report
All my concerns have been addressed. I recommend this paper for publication.